# Fast and Reliable Burst Data Transmission for Backscatter Communications

**DOI:** 10.3390/s19245418

**Published:** 2019-12-09

**Authors:** Jumin Zhao, Xiaojuan Liu, Dengao Li

**Affiliations:** 1College of Information and Computer, Taiyuan University of Technology, Taiyuan 030024, China; liuxiaojuan0146@sina.com; 2Technology Research Center of Spatial Information Network Engineering of Shanxi, Taiyuan 030024, China; lidengao@tyut.edu.cn; 3College of Data Science, Taiyuan University of Technology, Taiyuan 030024, China

**Keywords:** computational RFID (CRFID), burst data transmission, backscatter communications, erasure coding, goodput optimization

## Abstract

Computational radio frequency identification (CRFID) sensors are able to transfer potentially large amounts of data to the reader in the radio frequency range. However, the existing EPC C1G2 protocol is inefficient when there are abundant critical and emergency data to be transmitted and cannot adapt to changing energy-harvesting and channel conditions. In this paper, we propose a fast and reliable method for burst data transmission by fragmenting large data packets into blocks and we introduce a burst transmission mechanism to optimize the EPC C1G2 communication procedure for burst transmission when there are critical and emergency data to be transmitted. In addition, we utilize erasure codes to reduce Acknowledgement (ACK) delay and to improve system reliability. Our results show that our proposed scheme significantly outperforms the current fixed frame length approach and the dynamic frame length and charging time adaptation scheme (DFCA) and that the goodput is close to the theoretically optimal value under different energy-harvesting and channel conditions.

## 1. Introduction

Currently, miniature and ultra-low power sensors are widely used in urban and indoor areas. They are attached to common items and logistics to be used to track these objects [1]. Such devices utilize energy in the environment or RF to power and communicate with RFID readers by backscattering for data transmission. Computational radio frequency identification (CRFID) is a kind of passive sensing node that captures RF energy [2] and presents a new frontier for distributed sensing [3]. CRFIDs follow the RFID backscatter communication mode, which obtains energy from the RF signal emitted by the reader and modulates the encoded data back into the reader in the received signal. CRFID sensors are very different from existing battery sensor platforms and commercial RFID systems. They rely entirely on the energy of the capacitor for continuous sensing and efficient backscatter communication for data transmission. Compared to commercial RFID systems, they are able to sense, calculate, and store not just able to simply identify. In recent years, CRFID sensor systems have received increasing attention because of their potential for battery-free permanent sensing [4,5,6]. Typical CRFID systems include the WISP (Wireless Identification and Sensing Platform) [7], jointly developed by Intel and the University of Washington, and the UMass Moo [8], developed by the University of Michigan on the basis of WISP.

In this paper, we focus on mobile CRFID [9] and consider CRFID to use the harvesting energy for burst data transmission. CRFIDs are very small and can be deployed in large numbers indoors on moving objects or on the human body. As they move near the reader, the buffered data are transmitted to the reader via backscatter communication, such as mobility health monitoring. With the widespread deployment of CRFIDs, sensing tasks are becoming more complex and the sensing data is increasing. In particular, when it needs to be transmitted quickly and efficiently for critical and emergency data [10], such as clinical monitoring and industrial gas leaks, current backscatter communication brings some challenges. First, commercial RFID systems follow the EPC C1G2 protocol [11], which is designed to read a small amount of data (identifier EPC) for a large number of tags but is less efficient in scenarios where a small number of CRFIDs need to transfer large amounts of buffered data. Second, the key parameter controlling the efficiency of the EPC C1G2 protocol is the size of the window Q value [12], which is set by the reader based on its estimated number of tags. Also, the QueryRep or QueryAdj command of the tag listening to the unresponsive slot generates a lot of overhead, resulting in wasted energy and time [13]. Third, mobility of CRFID sensors results in dynamic changes in both energy-harvesting and channel conditions [14]. When the energy-harvesting conditions are poor, CRFID needs a long sleep charging time to reach the working voltage; when the channel quality is poor, bit errors easily occur, resulting in data frames being discarded, and CRFID needs to be recharged again, which leads to low goodput.

In response to the above difficulties, the following are the chief contributions of our work:We propose a method for burst data transmission by fragmenting large data packets into blocks. Then, we dynamically adjust the frame length of every block through an online adjustment strategy at runtime by the feedback of the reader.We introduce a burst transmission mechanism. The core idea is to let a tag occupy all time slots for burst transmission when there are critical and emergency data to be transmitted, which reduces the idle time slot.We utilize erasure codes to reduce acknowledgement frame waiting delay and to avoid retransmission overhead, which improves system robustness and reliability.Under specific energy-harvesting and channel conditions, our proposed scheme is much better than the EPC C1G2 protocol fixed frame length approach and DFCA scheme, and the performance can converge to near optimal. When there are abundant critical and emergency data to be transmitted, our scheme can achieve fast and reliable bulk burst data transmission.

The rest of the paper is as follows: We discuss related work in Section 2. In Section 3, we discuss the core challenges that need to be addressed to improve the backscatter system efficiency. In Section 4, we describe the CRFID operating procedure optimization by introducing a burst transmission mechanism and erasure coding. The design of the burst data transmission scheme is presented in Section 5, while the performance evaluation results are given in Section 6. Finally, we conclude the paper in Section 7.

## 2. Related Work

As “smart” transponders, UHF-passive RFID tags enhanced with computational and sensing capabilities and CRFIDs are qualified to become fully fledged components of the Internet of Things (IoT). Recent advances in smart systems driven by IoT technologies have opened up great opportunities for the development of backscatter communications. To enable RFIDs to be accessible from/to communicate with any other networked devices in the Internet, Reference [15] made information accessible through IoT6 architecture integration. This method only allows access to the data generated by the RFID system but does not allow the use of RFIDs for real-time communication. In order to extend the Internet Protocol Version 6 (IPv6) to the RFID world, transparent agents are used in References [16,17] to actively manage the resources and operations of tags. Reference [18] designed a hybrid sensor network medical system integrated with the Wireless Sensor Networks (WSN) and RFID nodes, which not only is suitable for identifying and tracking patients, caregivers, and biomedical equipment in hospitals but also provides remote monitoring and emergency management through three-axis acceleration. As far as the author knows, none of these studies involved the processing of emergency data generated by burst tags to make it quickly, efficiently, and reliably transmitted to the reader. In References [19,20], the Wi-Fi or bluetooth signal sent by the reader is used as the tag excitation signal, which greatly improves the system throughput and transmission distance. However, the self-interference problem of the signal is difficult to solve and it is not compatible with the existing EPC Gen2 protocol. Therefore, our research focuses on using the existing protocol or its improved protocol to improve the backscatter communication goodput.

In view of the problems for backscatter communications described in Section 1, References [21,22,23] designed a channel monitoring algorithm to adaptively adjust the data transmission rate to optimize the goodput by estimating the packet loss rate and the received signal strength indication (RSSI). QuarkNet [24] cuts data frames into very small data units to accommodate extremely poor environments. However, this work does not consider the transmission overheads such as the frame header overhead. MementOS [25] and Dewdrop [26] solve the problem of insufficient energy in different ways, but they are difficult to achieve in practice. Buzz [27] uses rateless code, but it must use a synchronous single-bit slot between nodes. FlipTracer [28] and BiGroup [29] studied the conflict of tags on the physical layer of RFID from two aspects, including constellation domain and time domain. Through parallel decoding of backscatter communication and assuming good channel conditions, they achieved large aggregated throughput. Our work focuses on the media access control (MAC) layer and differs in that we dynamically adjust frame length and coding redundancy under specific energy-harvesting and channel conditions while taking into account that transmission overhead and goodput can converge to near optimal. 

The dynamic adjustment of frame length and erasure code technology has been proven by many literatures to effectively enhance transmission reliability and system goodput under adverse channel conditions. In Reference [30], they proposed a dynamic segmentation scheme to enhance goodput in a time-varying wireless environment. Similar techniques have also been applied to WSN. In Reference [31], Dong et al. proposed a dynamic frame length control strategy for sensor networks, reducing communication overheads and improving energy utilization. The use of erasure coding in Reference [32,33] is used for data transmission in multi-hop of wireless sensor network (WSN) to improve network throughput, energy efficiency, and high reliability while greatly reducing end-to-end delay. Unlike the active radio communication scenarios described above, the good performance of backscatter communication is affected by channel conditions and energy-harvesting conditions. Our work is designed to maximize goodput and to adapt to energy-harvesting and channel conditions. 

## 3. Challenge and Motivation

In this section, we investigate the fundamental factors underlying the poor performance of backscatter communication and describe the main challenges that need to be addressed to improve the backscatter system efficiency.

### 3.1. Challenge 1: Variable Energy per Transmission

As shown in Figure 1, the CRFID node operates in a series of charging and discharging processes, which cannot work continuously due to too little energy. The device collects energy and charges a small energy store during a short period of sleep time and then wakes up and discharges to send packets.

Why is the energy available in each discharge cycle difficult to predict? First, if the energy-harvesting condition is too low, the efficiency of storing energy into the capacitor is low [24]. Therefore, the maximum energy that can be accumulated depends on the current harvest condition. Second, the RF energy collected by the node depends on how much energy the reader outputs. When the reader is communicating, the harvest rate of each node is constantly changing. Third, the node needs to use an analog-to-digital conversion (ADC) to measure the energy level. Each ADC operation consumes 327 µJ on the WISP platform, which is equivalent to the energy budget for transmitting 27-bit data. This cost is too much on a micro-powered platform.

Therefore, we need to adjust the length of the transmitted data according to the current energy environment. When the energy level is low, we need to shorten the frame length to adapt, but this usually reduces the goodput which is affected by the overhead of each transmission, including preamble, header, and hardware. In order to optimize goodput at the same time, it is important to transfer the data as much as possible given the available energy. Therefore, the problem faced by the node is that it needs to shorten its transmission frame under poor harvest conditions and to scale up to increase goodput when condition permits.

### 3.2. Challenge 2: Variable Harvesting Rate

The energy-harvesting rate has a significant impact on communication goodput, since higher harvesting rate means that more energy can be used for data transfer. Figure 2 shows the trend of energy-harvesting rate for both theoretical and empirical measurements as the sleep time between transmissions. One might expect to collect more energy by increasing the charging time. However, for longer sleep durations, the energy-harvesting rate drops to zero.

Next, we will explain this phenomenon by looking at how capacitors buffer energy. 

The charging process of the capacitor can be described the formula of voltage variation across the capacitor:(1)$$V=V_{\max}(1-e^{-t_{s}/\tau })$$
where $t_{s}$ is the sleep time, τ is the RC circuit time constant, and $V_{\max}$ is the maximum voltage achievable by the capacitor under the current energy-harvesting conditions. The energy-harvesting rate follows the following formula:(2)$$H=C\times V_{\max}^{2}\times \tau^{-1}(1-e^{-t_{s}/\tau })e^{-t_{s}/\tau }$$

When the harvesting conditions are constant (i.e., $V_{\max}$ and τ are fixed), H is a concave function of $t_{s}$. When the energy-harvesting condition changes, both $V_{\max}$ and τ change, so the optimal operating point also changes. When the tag’s capacitor stores more energy than the threshold, the energy-harvesting rate drops sharply from a high level to near zero. This means that, after getting enough energy, if the tag does not perform the task immediately, the energy-harvesting rate will drop sharply. Therefore, in order to optimize goodput, it is important to adapt to current energy-harvesting conditions and to keep track of the maximum harvesting points.

## 4. CRFID Operating Procedure Optimization

In this section, we first describe the optimized CRFID operating procedure. Then, we analyze goodput optimization problem by controlling the optimal number of transmission frames and sleep time of CRFID according to the current energy harvesting.

### 4.1. CRFID Operating Procedure

We introduce a burst transmission mechanism to optimize the EPC C1G2 communication procedure for bulk transmission. The core idea is fragmenting large data packets into blocks and letting a tag occupy all time slots for burst transmission when there are data to be transmitted. In addition, in order to deal with the error of data frames, we introduce erasure codes to improve system reliability and to reduce ACK delay.

An overview of erasure coding is given in Figure 3. The node encodes N source data frames that are required to transmit into N+M frames through the XOR of several random source data frames, where M is the number of redundant frames [34]. Each frame is sent only once, and the reader can decode and restore the original data by successfully receiving any N frames. Thus, there is no need to acknowledge each frame for the reader, which reduces ACK delay. If the number of errors in the data frame is greater than M+1, it means the data restoration failed.

The improved communication procedure is shown in Figure 4. The Query and QueryRep commands are used to initialize data requests and to set related parameters. The RN16 contains a 16-bit random number generated by the CRFID. After being sent to the reader, the reader returns it as a field of the ACK to the tag to indicate agreeing to the data transmission request of the tag. Then, the tag selects the optimal sleep time and the number of transmitted data frames according to the current energy-harvesting condition for burst transmission. Then, the reader and CRFID repeat the above duty cycle operation until all data in the current round have been sent or wait until the reader returns a QueryRep command for the next round of sending request. 

### 4.2. Goodput Analysis and Optimization

We have already known that longer sleep time does not necessarily result in higher goodput when energy-harvesting conditions are poor. So how do we improve goodput by adapting to current energy-harvesting conditions? By continuously tracking the maximum harvest point, as shown in Figure 2, we can obtain the optimal sleep time and energy-harvesting rate, calculate the RF energy captured by the CRFID node, and further derive the optimal number of transmission frames under the current condition. Here, we use the gradient descent algorithm to approximate the optimal value.

*Sleep time adaptation*: As can be seen from Figure 2, the energy-harvesting rate curve is a concave function of sleep time (under specific harvesting conditions). A fast and efficient way to converge to the optimal value of the concave function is used by the gradient descent algorithm. The gradient descent algorithm works as follows. We initialize the sleep time and calculate the gradient at this time. Then, we look for the direction of the positive gradient and move a certain step size in this direction. We repeat this iteration until the difference in energy-harvesting rate between the two iterations is small enough to indicate that the local optimum has been reached. If the step size is too small, the convergence may be too slow, and if the step size is too large, convergence cannot be guaranteed. Therefore, we should choose the appropriate step size according to the gradient. In addition, if the harvest conditions change, the curve will change. Our gradient-based sleep time adaption algorithm runs continuously once it converges to the optimal value. It periodically checks the gradient under the current optimal conditions and moves along a positive gradient as the optimal harvest rate changes. In this way, we get the optimal harvesting rate and sleep time at different energy-harvesting conditions.

*Optimal number of transmitted frames*: According to the above analysis, at the specific energy-harvesting condition, when the sleep time is $t_{o}$, the energy-harvesting rate is the maximum $H_{\max}$. The radio frequency energy E(t) captured by the CRFID at this time is as follows:(3)$$E(t)=\int_{0}^{t_{o}}C\times V_{\begin{array}{l} \max \end{array}}^{2}\times \tau^{-1}(1-e^{-t_{s}/t})e^{-t_{s}/t}dt_{s}$$

Suppose that the average energy cost by a node to transmit or receive unit-bit data is $e_{bit}$ and that the lengths of the data frame payload and the header (including the FCS field) are represented by $l_{p}$ and $l_{h}$, respectively. The energy cost required to transmit one frame is thus as follows:(4)$$E_{frame}=e_{bit}\times (l_{p}+l_{h})$$

The number of data frames for burst transmission in the kth duty cycle is represented by $n_{k}$. Since the captured energy cannot be less than the energy required to transmit the frame, there is the following inequation.

(5)$$E(t_{k})\geq n_{k}E_{frame}$$

According to Equation (5), the number of burst transmission data frames in the kth duty cycle satisfies the following inequation:(6)$$n_{k}\leq \min\left\{\left\lfloor \frac{E(t_{k})}{E_{frame}}\right\rfloor ,N+M-\sum_{i=1}^{k-1}n_{i}\right\}$$
where ⌊⌋ represents rounding down and $\sum_{i=1}^{k-1}n_{i}$ represents the total number of burst data frames of the previous *k*−1 duty cycles.

## 5. Burst Data Transmission Scheme

In this section, we present the design of our scheme by fragmenting large data packets into blocks. The major goal of our scheme is to achieve fast and reliable burst data transmission and goodput optimization for backscatter communications under dynamic harvesting and channel conditions. 

### 5.1. Design

The main idea of our solution is to fragment large data packets into blocks and to adjust the frame length and coding redundancy according to the goodput feedback of the reader at runtime, meanwhile controlling the sleep time and the number of transmitted frames to improve the goodput.

Our scheme works as follows: CRFID nodes periodically track the voltage of the capacitor and estimate $U_{\max}$ and τ. When the CRFID node moves to the vicinity of the reader and there are data to be sent in the buffer, N source data frames of length $l_{p}$ are extracted from the buffer and encoded into N+M frames, and the initial frame length is set to 16 bits. It communicates with the reader following the procedure described in Figure 4. The reader monitors the received frame. When a round of data transmission is completed, the reader obtains the time spent and goodput and compares the current round goodput with the previous round. If the current round goodput is higher than the previous round goodput, the ACK is returned to inform the CRFID node to increase the length of the transmitted frame; otherwise, the frame length is reduced, and the unit of increasing or decreasing the frame length is 8 bits.

To avoid frequent changes to the frame length, the adjustment parameter θ(0≤θ≤0.1) is introduced. If the goodput improvement or degradation does not exceed θ, the frame length is not changed. At the same time, after receiving the frame length specified by the reader, the CRFID node selects the number of frames to be transmitted in the current duty cycle according to the current energy-harvesting condition. Then, it enters the sleep state after the transmission is completed and waits for the next working cycle until all data of the current round have been sent or waits until the reader sends a QueryRep frame. 

After one round of transmission, the reader counts the number of redundant frames M actually transmitted in the case of successful transmission of the current round. If the transmission of the current round fails, the number of redundant frames in the next round is updated according to Equation (7).
(7)$$\begin{array}{l} M_{\mathrm{next}}=\left\lceil (1-\alpha )\times M_{\mathrm{avg}}+\alpha \times M_{\mathrm{cur}}+M_{\min}\right\rceil \\ \alpha \in (0,1),M_{\mathrm{next}}\in (M_{\min},M_{\max}) \end{array}$$
where $M_{\mathrm{avg}}$ denotes the number of historical average transmitted redundant frames, $M_{\mathrm{cur}}$ denotes the number of redundant frames transmitted in the current round, $M_{\min}$ and $M_{\max}$ respectively represent the minimum and maximum values of the number of redundant frames, α is the weight coefficient, and ⌈⌉ represents rounding up.

Specifically, the CRFID node operates as follows:

Step 1: If the CRFID receives Query or QueryRep command from the reader, an RN16 is returned.

Step 2: If an ACK of the reader is received and the ACK contains the same 16-bit random number as the RN16, N source data frames of length $l_{p}$ are extracted from the buffer. The N source data frames are encoded into N+M frames, where the values of M and $l_{p}$ are obtained from the information carried in the ACK.

Step 3: The capacitance voltage is tracked. $U_{\max}$ and τ are estimated at the initial moment of the duty cycle k.

Step 4: Calculate the optimal number of burst transmission frames $n_{k}$ of the duty cycle k according to Equation (6) and control the optimal sleep time $t_{o}$.

Step 5: After sleep $t_{o}$, $n_{k}$ frames are continuously transmitted. If the reader receives a QueryRep in the current duty cycle, the CRFID immediately gives up the transmission and runs the first step; otherwise, Step 3 is run until all N+M frames have been transmitted.

At the same time, the reader operates as follows:

Step 1: The reader sends a Query to request data and returns an ACK if it receives the RN16 of the CRFID. The ACK includes the same random number as in the RN16 and the frame length $l_{p}$ used by the CRFID for subsequent transmission and the number of redundant frames M. The initial value of $l_{p}$ is 16 bits and the initial value of M is $M_{\min}$.

Step 2: The number of buffered data V, the number of correctly received frames R, and the number of incorrect frames F are all set to zero.

Step 3: If the data frame of the CRFID is received, let $V=V+l_{p}$. If the data frame is correct, let R=R+1; otherwise, let F=F+1. Further, judge that, if R=N, N source data is restored and run Step 4. If F=M+1, the round transmission fails and run Step 4; otherwise, run Step 3.

Step 4: Calculate the current goodput G=V/Δt, and Δt is the transmission time of the current round measured by the reader. If $G>G^{\prime }(1+\theta )$ and $l_{p}<l_{p}^{\max}$, let $l_{p}=l_{p}+8$, and if $G<G^{\prime }(1+\theta )$ and $l_{p}>l_{p}^{\min}$, let $l_{p}=l_{p}-8$; otherwise do not change the data frame length, where $G^{\prime }$ is the goodput of the last round. Calculate the number of redundant frames M for the next round according to Equation (7).

Step 5: The reader sends a QueryRep command to request data and returns an ACK frame if receiving the RN16 frame of the CRFID. The ACK contains the same random number as in the RN16 and the transmitting information such as the frame length and the number of redundant frames calculated in Step 4. Then, run Step 2.

Pseudo codes of the CRFID and the reader operation procedure are shown in Algorithm 1 and Algorithm 2, respectively.

**Algorithm 1** The computational radio frequency identification (CRFID) operation procedure. 1: **while** the buffer id not empty **do** 2: **if** receive Query or QueryRep command **then** 3:  return RN16 4: **end** 5: **if** receive the ACK containing the same RN16 **then** 6:   extract N data frames of length lp and encode into N+M (lp and M are carried in the ACK) 7: **end** 8: **repeat** 9:  track the capacitance voltage and obtain Umax and τ, and sleep for a period of to10:  compute the optimal number of transmission frames nk via Equation (6); then, transmit it to the reader11: **until** all N+M frames have been transmitted12: **end**

**Algorithm 2** The reader operation procedure. 1: Initialize $l_{p}$ = 16 bits, M = $M_{\min}$, V = 0, R = 0, F = 0 2: send the Query or QueryRep command 3: **if** receive the RN16 from CRFID **then** 4:   return an ACK with the same RN16 and $l_{p}$ and M 5: **end** 6: **if** receive the frame from CRFID 7:   **if** the frame is correct **then** 8:   $V=V+l_{p}$; R=R+1 9:   **else**
F=F+110:  **end**11:  **if**
R=N or F=M+1
**then**12:  compute G=V/Δt, where Δt is the transmission time of the current round13:   **if**
G>G′(1+θ) and lp<lpmax
**then**14:     lp=lp+815:   **else if**
G<G′(1+θ) and lp>lpmin
**then**16:     lp=lp−817:  **end**18: **end**19: compute the number of redundant frames M via Equation (7) for the next round20: **end**

### 5.2. Frame Format

The data frame format used is similar to the data frame format of the EPC C1G2 protocol, as shown in Figure 5. The data payload length in the EPC C1G2 protocol is defined by the EPC length field in the Protocol Control (PC) field, which ranges from 16 to 496 bits and is an integer multiple of 16 bits. Since the unit of increasing or decreasing the frame length in our scheme is 8 bits, the field is modified so that each bit represents 8 bits. Considering that the device only supports 96 bits, the maximum value of the field is 96 bits, that is, 01100. The ACK format of EPC C1G2 consists of 2 bits of command bits and 16-bit RN. However, in our proposed scheme, the frame length and the number of redundant frames need to be adjusted according to energy-harvesting and channel conditions, which are fed back to the CRFID by the reader through the ACK. Therefore, two additional 5-bit fields are added in the ACK, respectively indicating the frame length and the number of redundant frames in the next round, and the ACK format is as shown in Figure 6.

## 6. Performance Evaluation

### 6.1. Platforms

The experimental platform is shown in Figure 7. We use the Universal Software Radio Peripheral (USRP) N210 software-defined radios and WISP as backscatter nodes for our instantiation of our scheme. The USRP N210 is equipped with an SBX40 daughter board that can be used as a detector and reader. We use the open source code written by Nikos on GitHub to use USRP as a reader. Based on the experimental conditions and design, the SBX-40 daughter board provides the multiple-input multiple-output (MIMO) function and provides 40-MHz bandwidth. The working frequency is 400 MHz to 4400 MHz. The platform used in the experiment is 64-bit Ubuntu 14.04, GNU Radio 3.7.4. The selected CRFID tag is WISP4.1 with dipole antenna, and the microcontroller is MSP430F2132. The commercial reader model used is the ImpinJ Speedway R420, which is connected to the Laird circularly polarized directional antenna S9028PCL. Up to 4 antennas can be connected at the same time, and the antenna gain is 9 dBi. Our goal in the evaluation is to demonstrate that our proposed scheme can significantly improve system goodput. 

### 6.2. Trimming Overheads

*Erasure coding implementations*: It is obvious that the redundancy introduced using erasure codes increases energy consumption as its number increases. However, traditional reliability improvement methods, such as data multiplexing or Automatic Repeat Requests (ARQ) [35], are too costly and even impossible to implement due to strict energy constraints and channel asymmetry. Erasure codes allow the reliability of data transmission to be increased by transmitting redundant data. In our work, we investigated the potential for communication using erasure codes and investigated the trade-offs between reliability and energy consumption. For our platforms with limited energy and size, we mainly focus on Cauchy-matrix-based and Vandermonde-matrix-based Reed–Solomon (RS) code through existing open source implementations.

As a benchmark experiment, we simulated three cases: the normal case, that is, the data packet without encoding; the full copy, that is, the data is sent with an exact copy; and the on-demand retransmission, that is, if the ACK command that the data has been successfully sent is not received, the data will be resent. We use simulation and energy measurement on real hardware platforms to describe the overhead of redundant packets by setting different coding rates *r* to evaluate these methods. Our results clearly show that erasure coding has the same or less overhead compared to traditional data replication and on-demand retransmission and can provide higher reliability in our scenario. When erasure coding is used, as the code overhead and data rate increase, we observe that the recovery rate increases steadily compared to the other two methods of recovering data. In particular, erasure codes based on Reed–Solomon clearly outperform simple data replication and ARQ, as shown in Figure 8. All in all, based on measurements on hardware platforms with very limited actual energy consumption, we can prove that the computational cost of encoding is feasible and we can also notice the energy consumption on the nodes.

*Probing energy state*: As mentioned earlier, the ADC costs too much energy and should be avoided when tracking the maximum energy-harvesting rate. In our work, instead of measuring the voltage on nodes, we use existing low watermark threshold detectors that already exist on such nodes. This type of detector is very common on passive sensor platforms to control the state of the tag, when it should sleep to avoid interruptions, and when it should wake up to continue operation. Therefore, our algorithm is interrupted when the voltage exceeds the threshold or when the voltage is below the threshold, and this information is used as a one-bit proxy for the actual voltage. The voltage threshold is chosen to be 2 V, which is slightly higher than the minimum voltage of 1.8 V required to operate the microcontroller. This information is entered into the sleep time tracker, which decides how long to wait after exceeding the threshold before starting the transmission. Compared with ADC, this method saves about 100× in energy.

### 6.3. Evaluation

To model the changing energy-harvesting and channel conditions, we evaluate the performance with different values of $E_{b}/N_{0}$ and different combinations of $U_{\max}$ and τ. $E_{b}/N_{0}$ is the signal-to-noise ratio (SNR) per bit. All the corresponding parameters are listed in Table 1.

We consider that the CRFID has 7400-bit data in buffer that need to be sent to the reader. Our proposed scheme is simulated to obtain the goodput performance and the corresponding frame length and number of redundant frames, which are compared with the theoretical optimal solution obtained from Reference [36]. The theoretical optimal value is used as a benchmark under specific energy-harvesting and channel conditions. We also evaluate the performance of our proposed scheme by the actual implementation and compared it to the fixed 96-bit frame length strategy adopted by EPC C1G2 and the DFCA scheme. In order to be fair, all solutions use the optimized communication procedure as shown in Figure 4.

Figure 9a shows the variation of the goodput of the four schemes as $E_{b}/N_{0}$ increases under the harvesting conditions of $U_{\max}=6$ V and τ=2. We clearly see that, in Figure 9a, as $E_{b}/N_{0}$ increases, system goodput gradually increases due to the decreasing the bit error rate (BER). However, as $E_{b}/N_{0}$ gets larger, the improvement of goodput is reduced and eventually converges to a fixed value. This is because the frame successful transmission rate eventually tends to 1 with the decreasing BER. The theoretical goodput in Figure 9a is larger than our scheme, the DFCA scheme, and the fixed frame length scheme, but the theoretical goodput depends on accurate estimation about channel-quality and harvesting condition, which is difficult to achieve in practice. The goodput achieved by our solution is very close to optimal.

Figure 9b,c shows the optimal frame length and the number of redundant frames used by the theoretical method and our proposed scheme, respectively, under $E_{b}/N_{0}$ corresponding to Figure 9a. It can be seen that, when $E_{b}/N_{0}$ is small, which means poor channel quality, the frame length used is small and the number of redundant frames is large. As $E_{b}/N_{0}$ increases, the frame length is larger and redundant frames are fewer. Our scheme is close to the theoretical value, but because our scheme uses the online measurement goodput feedback to the node to change the corresponding frame length and the number of redundant frames, the curve has certain fluctuations. 

Figure 10 shows the goodput, frame length, and number of redundant frames for several scenarios under energy-harvesting conditions of $U_{\max}=5$ V and τ=3. Compared with Figure 9, the maximum chargeable voltage of the CRFID node is reduced to 5 V and the RC circuit time constant τ is set to 3, which reduces the energy-harvesting conditions. Overall, trend of goodput variation in Figure 10a is the same as in Figure 9a, but the overall goodput is smaller than Figure 9a due to poor energy-harvesting conditions. As can be seen from Figure 10a,b, increasing the frame length does not increase the goodput indefinitely. This is because charging time becomes a major factor that mainly affects goodput in the case of better channel conditions and poor energy-harvesting conditions. Therefore, under this condition, obtaining a larger transmission frame length at the cost of a longer charging time may cause a decrease in goodput.

Figure 11 shows the communication energy consumption of our scheme and fixed frame length strategy under $U_{\max}=5$ V and τ=3. It can be seen from Figure 11 that, in the case of low $E_{b}/N_{0}$, our scheme consumes more energy than the fixed frame length strategy, while in the case of high $E_{b}/N_{0}$, our scheme consumes less energy. This is because, when $E_{b}/N_{0}$ is low, our scheme guarantees a successful packet-receiving rate by increasing the number of redundant packets, so that the energy consumption is increased. Also, when the signal-to-noise ratio is better, the number of redundant packets is reduced and our scheme eliminates the need to return an acknowledgment to per frame to save energy. It is worth mentioning that our paper does not consider the calculation energy consumption because it is of a smaller order of magnitude compared to communication energy consumption. Moreover, the core algorithm of the method used in this paper runs on the reader side, and the calculation amount of the CRFID is small.

## 7. Conclusions

In this paper, we achieve fast and reliable bulk burst data transmission by optimizing goodput based on burst transmission when there are critical and emergency data to be transmitted. First, we optimize the EPC C1G2 protocol by introducing burst transmission mechanism and erasure codes and control the optimal number of transmission frames and sleep time of CRFID according to the current energy-harvesting and channel conditions. Then, we fragment large data packets into blocks and design an online adjustment strategy to adjust the frame length and coding redundancy dynamically by the feedback of the reader at runtime. Our results show that our proposed scheme significantly outperforms the current fixed frame length approach and the DFCA scheme, and the goodput is close to the theoretically optimal value under different energy-harvesting and channel conditions.

Our proposed scheme in this paper enables passive sensing communication to adapt to dynamic energy-harvesting and channel conditions and to facilitate their application in the field of mobile sensing and pervasive computing. Future research work will consider the issue of data frame retransmission and how to deal with collisions and to ensure data transmission priority to further improve system performance in multi-tag scenario. In addition, this method can be combined with ambient back-scattering technology to make use of all the energy available in the environment such as temperature difference energy supply, mechanical vibration energy, etc. to improve the intelligence of smart devices and to establish a true smart system. That is to say, in the near future, data communication will not be restricted by the outside world, and it can flexibly realize the conversion between different protocols, thereby achieving barrier-free communication between various IoT devices.

## Figures and Tables

**Figure 1 sensors-19-05418-f001:**
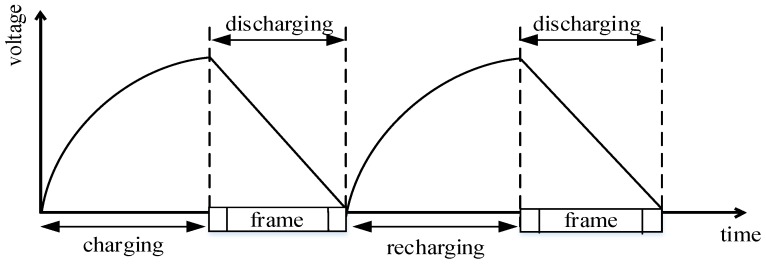
Energy-harvesting process.

**Figure 2 sensors-19-05418-f002:**
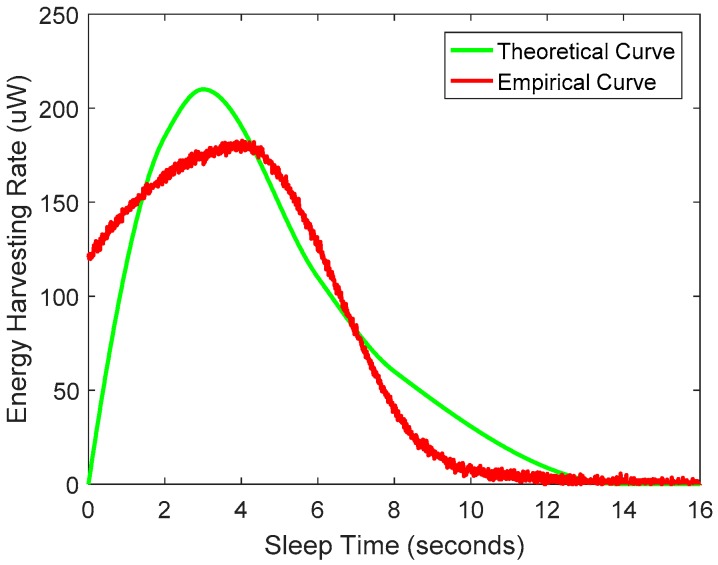
Energy-harvesting rate curve.

**Figure 3 sensors-19-05418-f003:**
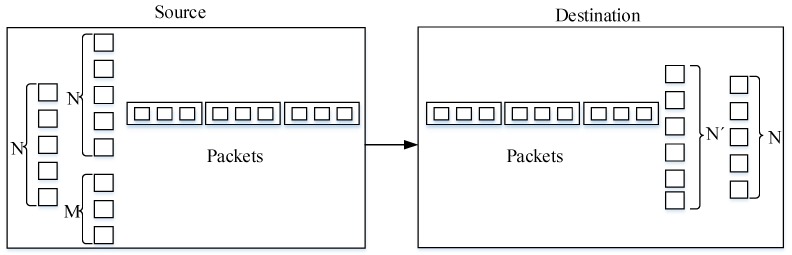
Overview of erasure coding.

**Figure 4 sensors-19-05418-f004:**
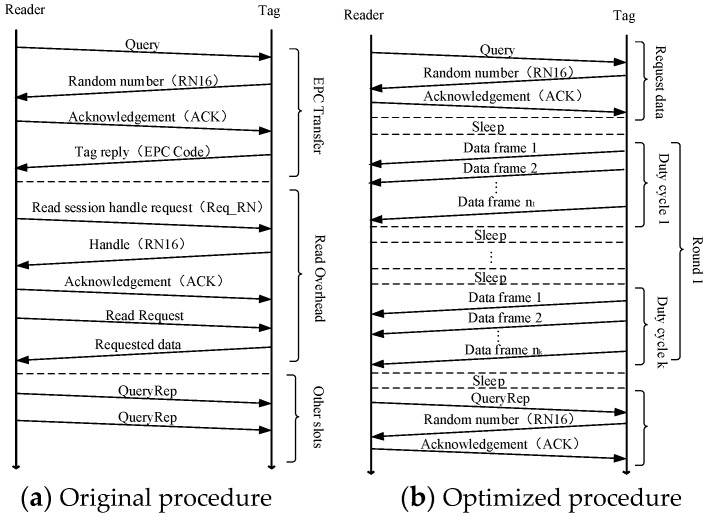
Operating procedure.

**Figure 5 sensors-19-05418-f005:**
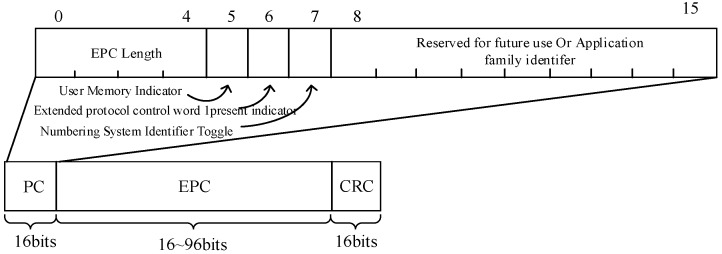
Format of the data frame.

**Figure 6 sensors-19-05418-f006:**
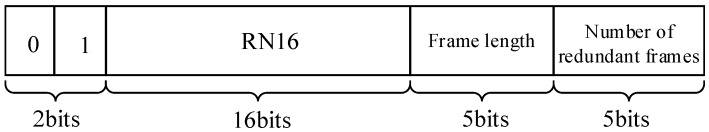
Format of the ACK message.

**Figure 7 sensors-19-05418-f007:**
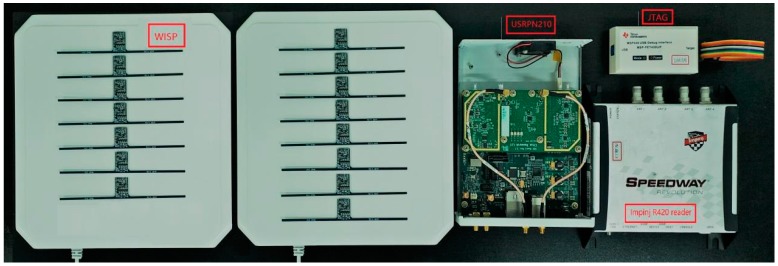
Experiment equipment.

**Figure 8 sensors-19-05418-f008:**
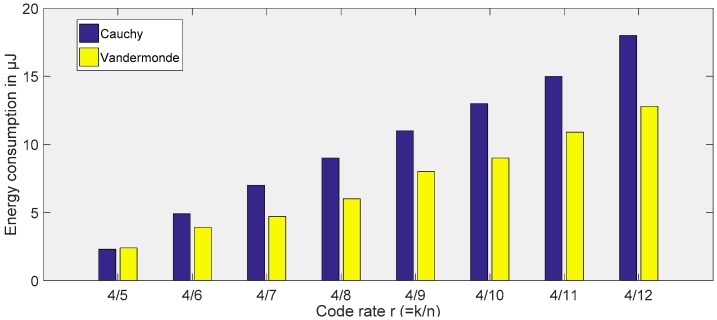
Energy consumption comparison between Cauchy- and Vandermonde-matrix–based Reed–Solomon (RS) code.

**Figure 9 sensors-19-05418-f009:**
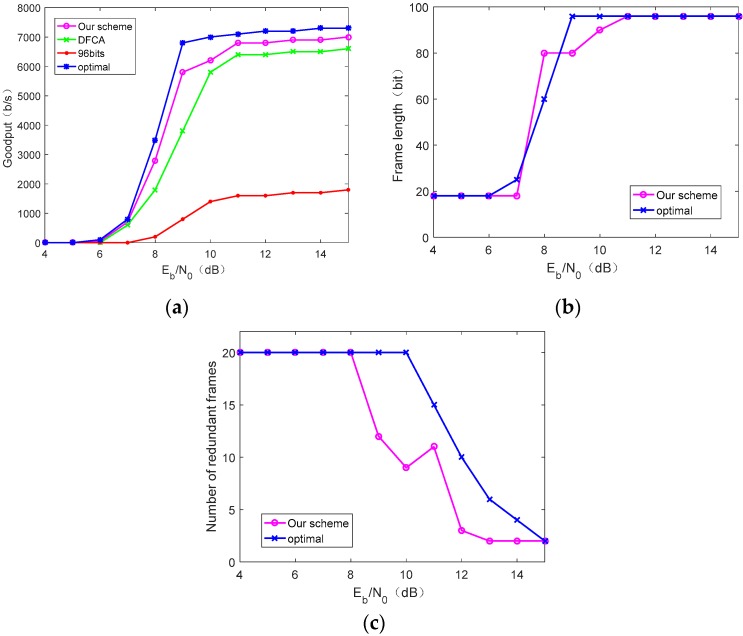
Performance comparison under the conditions $U_{\max}=6$ V and τ=2: (**a**) The goodput performance under different values of $E_{b}/N_{0}$, (**b**) frame length under different values of $E_{b}/N_{0}$, and (**c**) number of redundant frames under different values of $E_{b}/N_{0}$.

**Figure 10 sensors-19-05418-f010:**
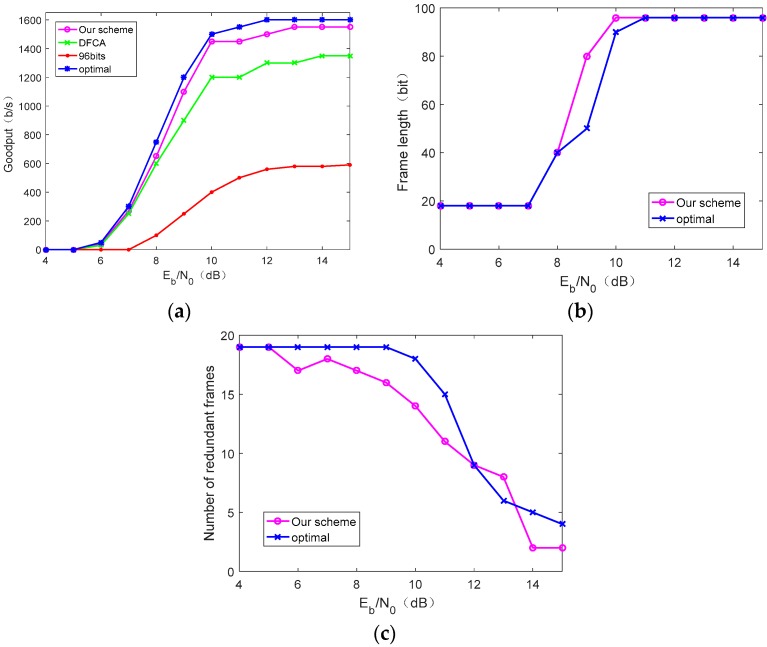
Performance comparison under the conditions $U_{\max}=5$ V and τ=3: (**a**) The goodput performance under different values of $E_{b}/N_{0}$, (**b**) frame length under different values of $E_{b}/N_{0}$, and (**c**) number of redundant frames under different values of $E_{b}/N_{0}$.

**Figure 11 sensors-19-05418-f011:**
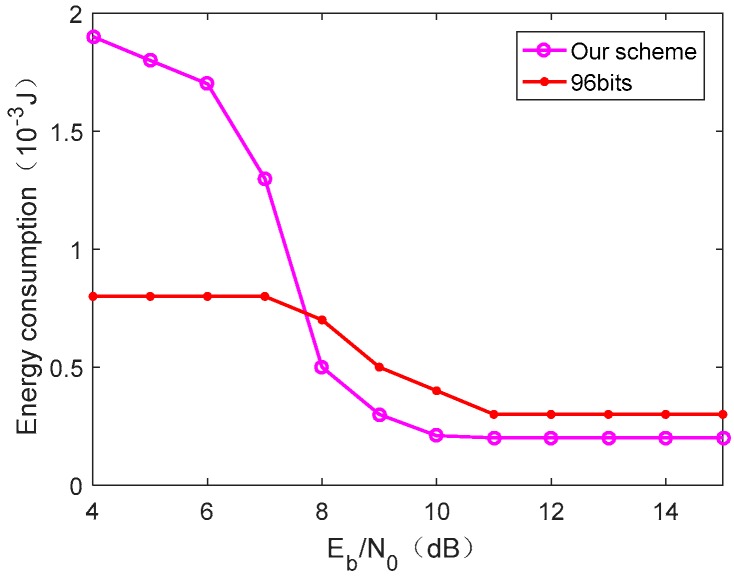
Energy consumption under the conditions $U_{\max}=5$ V and τ=3.

**Table 1 sensors-19-05418-t001:** Corresponding parameters setting.

Parameters	Values
payload length $l_{p}$ (bit)	16~96
PC and CRC length $l_{h}$ (bit)	32
RC constant τ	2, 3
maximum voltage $U_{\max}$ (V)	6, 5
weight coefficient α	0.35
minimum redundant frames $M_{\min}$	1
maximum redundant frames $M_{\max}$	20

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
