# Peer review of "Fast and Reliable Burst Data Transmission for Backscatter Communications"

_sensors, 2019, doi:10.3390/s19245418_

Round 1
Reviewer 1 Report
The document is well written, which makes reading easier.
Good contribution, clearly defined in page 2 (chief contributions of our work). It indicates a set of devices, but would it be possible to integrate others and, in this case, it would need to include a definition or design of general architecture?
Good experimentation set included in the experimentation section.
Most of the references in section 2. "Related work" are not very current. Many are from the years 2011-2012 and there are no later than 2014. I am sure you can find more current works in the last 5 years.
The paragraph of the future lines seems rather short. I suggest that it be extended, indicating in which application domains it could be incorporated. The question is whether they are thinking about including devices, such as sensors that are gathering information. Then why the experiment is carried out through a simulation instead of doing it on a real application scenario or real environment?
Author Response
Response to Reviewer 1 Comments
The document is well written, which makes reading easier.
Good contribution, clearly defined in page 2 (chief contributions of our work). It indicates a set of devices, but would it be possible to integrate others and, in this case, it would need to include a definition or design of general architecture?
Response 1: I'm so sorry because I may not fully understand what you mean. From my personal understanding, the reply is as follows:
The main equipment used in our experiments includes passive-aware computing tags and a special reader. The reader is a software-defined radio (SDR) reader built using the USRP, which requires some parameter settings, such as channel and modulation. And, the method in this paper can be used on any passive-aware computing tags that follow the standard EPC Gen2 protocol, such as the UMass Moo developed by the University of Michigan on the basis of WISP.
Good experimentation set included in the experimentation section.
Most of the references in section 2. "Related work" are not very current. Many are from the years 2011-2012 and there are no later than 2014. I am sure you can find more current works in the last 5 years.
Response 2: Thanks for your opinions, a wider span of literature and current researches in the last 5 years on backscatter communication have been added in my manuscript. Please check it!
The paragraph of the future lines seems rather short. I suggest that it be extended, indicating in which application domains it could be incorporated. The question is whether they are thinking about including devices, such as sensors that are gathering information.
Response 3: Thanks for your opinions, we have expanded our future work.
This method can be combined with ambient back-scattering technology to make use of all the energy available in the environment such as temperature difference energy supply, mechanical vibration energy, etc. to improve the intelligence of smart devices and establish a true smart system. That is to say, in the near future, data communication will not be restricted by the outside world, and it can flexibly realize the conversion between different protocols, thereby achieving barrier-free communication between various IoT devices.
The above is also supplemented in our manuscript, please check it!
Then why the experiment is carried out through a simulation instead of doing it on a real application scenario or real environment?
Response 4: I am so sorry for the misunderstanding caused by the mistakes I made. In the manuscript, we have checked for incorrect expressions. Please check it!
Our results are obtained by the actual implementation on USRP as a special reader and the CRFID tag. A software-defined radio (SDR) reader built using the USRP can be used to explore protocol modifications. By carrying different daughter boards to make them work at different channel frequencies, the MAC layer and physical layer protocols of the passive sensing system can be modified as needed through USRP.

Reviewer 2 Report
Authors propose a method for burst data transmission by fragmenting large data packets into blocks.
The manuscript needs a major revision prior to be processed further.
In the following, the comments that arise from the analysis of the manuscript are reported.
1. Related works should cover a wider span of literature, as other examples of packing larger data packets using EPC C1G2 standard protocols. Some examples:
IPv6 addressing proxy: Mapping native addressing from legacy technologies and devices to the internet of things (IPv6)
(2013) Sensors (Switzerland), 13 (5), pp. 6687-6712.
An IoT-Aware Architecture for Smart Healthcare Systems
(2015) IEEE Internet of Things Journal, 2 (6), art. no. 7070665, pp. 515-526.
6lo-RFID: A framework for full integration of smart UHF RFID tags into the internet of things
(2017) IEEE Network, 31 (5), art. no. 8053480, pp. 66-73.
Some references about hardware alternatives to WISP and the analysis of the impact of the implementation of the proposed method on such different architectures are also suggested.
2. Section 3 - Challenge 2
The authors seem to doubt that more energy is collected as charging times increase. It is not clear, notwithstanding the goodput could decrease for longer charging times.
3. The equation (1) is the voltage variation across the capacitor, not the charging equation.
4. Describe the model used for equation (2)
5. Provide further details on the erasure coding method. Is the overhead of data sent worth the result?
6. Is the method presented in Fig. 4 using EPC C1G2 standard functions? Does it require a special hardware/reader to be implemented?
7. The analysis of paragraph 4.2 does not take into considerations several but crucial aspects. The operations requested to CRFID to implement the gradient descent algorithm should require the usage of ADC, several micro controller instructions to calculate the values, canceling the added value of the algorithm implementation from an energy saving point of view. The analysis must be justified providing the actual power consumption of the operations necessary for the execution of the instructions.
8. The authors provide a detailed description of the setup for the performance evaluation, but it is not clear if the results are obtained by simulations on USRP or by the actual implementation of the hardware for both the CRFID tag and the reader.
For every comment, please provide both manuscript modification and reviewer response.
Author Response
Response to Reviewer 2 Comments
Related works should cover a wider span of literature, as other examples of packing larger data packets using EPC C1G2 standard protocols. Some examples:
IPv6 addressing proxy: Mapping native addressing from legacy technologies and devices to the internet of things (IPv6) (2013) Sensors (Switzerland), 13 (5), pp. 6687-6712.
An IoT-Aware Architecture for Smart Healthcare Systems (2015) IEEE Internet of Things Journal, 2 (6), art. no. 7070665, pp. 515-526.
6lo-RFID: A framework for full integration of smart UHF RFID tags into the internet of things (2017) IEEE Network, 31 (5), art. no. 8053480, pp. 66-73.
Some references about hardware alternatives to WISP and the analysis of the impact of the implementation of the proposed method on such different architectures are also suggested.
Response 1: Thanks for your opinions, a wider span of literature has been added in my manuscript, which contains the above examples and recent researches on backscatter communication in the Internet of Things. Please check it!
The method in this paper can be used on any passive-aware computing tags that follow the standard EPC Gen2 protocol, such as the UMass Moo developed by the University of Michigan on the basis of WISP.
Section 3 - Challenge 2
The authors seem to doubt that more energy is collected as charging times increase. It is not clear, notwithstanding the goodput could decrease for longer charging times.
Response 2: The energy harvesting rate has a significant impact on communication goodput, since higher harvesting rate means that more energy can be used for data transfer. One might expect to collect more energy by increasing the charging time, however, for longer sleep durations, the energy harvesting rate drops to zero. This is because when the tag's capacitor stores more energy than the threshold, the energy harvesting rate drops sharply from a high level to near zero. This means that after getting enough energy, if the tag does not perform the task immediately, the energy harvesting rate will drop sharply. Thus, it is not advisable to trade in longer charging time for greater goodput.
The corresponding content has been modified in my manuscript, please check it!
The equation (1) is the voltage variation across the capacitor, not the charging equation.
Response 3: Thanks for your opinions, we have modified the description of equation (1), please check it!
Describe the model used for equation (2)
Response 4: The figure below is a circuit diagram of the whole tag, where the capacitor in the tag and sensors is energy storage capacitor. Under certain energy harvesting conditions, the distance d between the reader and WISP is fixed, and Vmax determined accordingly. Then the tag has a fixed energy harvesting rate H.
Where C is the value of the capacitance, ts is the sleep time, is the RC circuit time constant, and Vmax is the maximum voltage achievable by the capacitor under the current energy harvesting conditions.
Circuit for the whole tag
Provide further details on the erasure coding method. Is the overhead of data sent worth the result?
Response 5: It is obvious that the redundancy introduced using erasure codes increases energy consumption as its number increases. However, traditional reliability improvement methods, such as data multiplexing or Automatic Repeat Requests (ARQ), are too costly and even impossible to implement due to strict energy constraints and channel asymmetry. Erasure codes allow the reliability of data transmission to be increased by transmitting redundant data. In our work, we investigated the potential for communication using erasure codes and investigated the trade-offs between reliability and energy consumption. For our platforms with limited energy and size, we mainly focus on Cauchy-matrix-based and Vandermonde-matrix-based Reed-Solomon (RS) code through existing open source implementations.
As a benchmark experiment, we simulated three cases: the normal case, that is, the data packet without encoding; the full copy, that is, the data is sent with an exact copy; the on-demand retransmission, that is, the ACK command that the data has been successfully received is not received, the data will be resent. We use simulation and energy measurement on real hardware platforms to describe the overhead of redundant packets by setting different coding rates r to evaluate these methods. Our results clearly show that erasure coding has the same or less overhead compared to traditional data replication and on-demand retransmission, and can provide higher reliability in our scenario. When erasure coding is used, as the code overhead and data rate increase, we observe that the recovery rate increases steadily compared to the other two methods of recovering data. In particular, erasure codes based on Reed-Solomon clearly outperform simple data replication and ARQ, as shown in Figure 8. All in all, based on measurements on hardware platforms with very limited actual energy consumption, we can prove that the computational cost of encoding is feasible, and we can also notice the energy consumption on the nodes.
Figure8. Energy consumption comparison
The above is also supplemented in our manuscript, please check it.
Is the method presented in Fig. 4 using EPC C1G2 standard functions? Does it require a special hardware/reader to be implemented?
Response 6: Fig.4 (b) is an improved protocol of the EPC C1G2 standard protocol (as shown in Figure 4 (a)), which can implement the functions of the standard protocol. It requires a special reader to be implemented. In this article, we use the USRP N210 software defined radios to replace the commercial reader- ImpinJ Speedway R420. The USRP N210 is equipped with a SBX40 daughter board that can be used as a detector and a special reader. We use the open source code written by Nikos on GitHub to use USRP as a reader.
The analysis of paragraph 4.2 does not take into considerations several but crucial aspects. The operations requested to CRFID to implement the gradient descent algorithm should require the usage of ADC, several micro controller instructions to calculate the values, canceling the added value of the algorithm implementation from an energy saving point of view. The analysis must be justified providing the actual power consumption of the operations necessary for the execution of the instructions.
Response 7: As mentioned in my manuscript, the ADC costs too much energy and should be avoided when tracking the maximum energy harvesting rate. In our work, instead of measuring the voltage on nodes, we use existing low watermark threshold detectors that already exist on such nodes. This type of detector is very common on passive sensor platforms to control the state of the tag, when it should sleep to avoid interruptions, and when it should wake up to continue operation. Therefore, our algorithm is interrupted when the voltage exceeds the threshold or when the voltage is below the threshold, and this information is used as a one-bit proxy for the actual voltage. The voltage threshold is chosen to be 2V, which is slightly higher than the minimum voltage of 1.8V required to operate the microcontroller. This information is entered into the sleep time tracker, which decides how long to wait after exceeding the threshold before starting the transmission. Compared with ADC, this method saves about 100 × in energy.
The above is also supplemented in our manuscript, please check it.
The authors provide a detailed description of the setup for the performance evaluation, but it is not clear if the results are obtained by simulations on USRP or by the actual implementation of the hardware for both the CRFID tag and the reader.
Response 8: Our results are obtained by the actual implementation on USRP as a special reader and the CRFID tag. Commercial readers are non-open source readers that are packaged and can be used directly for measurement and experimentation. A software-defined radio (SDR) reader built using the USRP can be used to explore protocol modifications. By carrying different daughter boards to make them work at different channel frequencies, the MAC layer and physical layer protocols of the passive sensing system can be modified as needed through USRP. The figure below shows the basic system framework for building a reader using USRP. The overall process is: USRP sender - matched filter - tag response gate - clock recovery - tag decoder - USRP receiver.
Basic framework of USRP reader architecture

Round 2
Reviewer 2 Report
The authors accepted my previous comments and suggestions.
The manuscript is now worth for publication in this journal.